# Chromosomal Microarray Analysis Identifies a Novel SALL1 Deletion, Supporting the Association of Haploinsufficiency with a Mild Phenotype of Townes–Brocks Syndrome

**DOI:** 10.3390/genes14020258

**Published:** 2023-01-19

**Authors:** Anna Maria Innoceta, Giulia Olivucci, Giulia Parmeggiani, Emanuela Scarano, Antonella Pragliola, Claudio Graziano

**Affiliations:** 1Medical Genetics Unit, AUSL Romagna, 47522 Cesena, Italy; 2Medical Genetics Unit, IRCCS Azienda Ospedaliero-Universitaria di Bologna, 40138 Bologna, Italy; 3Rare Diseases Unit, Department of Pediatrics, IRCCS Azienda Ospedaliero-Universitaria di Bologna, 40138 Bologna, Italy

**Keywords:** SALL1, Townes–Brocks syndrome, genotype–phenotype correlations, nonsense-mediated decay

## Abstract

SALL1 heterozygous pathogenic variants cause Townes–Brocks syndrome (TBS), a condition with variable clinical presentation. The main features are a stenotic or imperforate anus, dysplastic ears, and thumb malformations, and other common concerns are hearing impairments, foot malformations, and renal and heart defects. Most of the pathogenic *SALL1* variants are nonsense and frameshift, likely escaping nonsense-mediated mRNA decay and causing disease via a dominant-negative mechanism. Haploinsufficiency may result in mild phenotypes, but only four families with distinct *SALL1* deletions have been reported to date, with a few more being of larger size and also affecting neighboring genes. We report on a family with autosomal dominant hearing impairment and mild anal and skeletal anomalies, in whom a novel 350 kb *SALL1* deletion, spanning exon 1 and the upstream region, was identified by array comparative genomic hybridization. We review the clinical findings of known individuals with *SALL1* deletions and point out that the overall phenotype is milder, especially when compared with individuals who carry the recurrent p.Arg276Ter mutation, but with a possible higher risk of developmental delay. Chromosomal microarray analysis is still a valuable tool in the identification of atypical/mild TBS cases, which are likely underestimated.

## 1. Introduction

The utility of chromosomal microarray analysis (CMA) for the detection of copy number variants (CNVs) in patients with neurodevelopmental disorders and/or congenital malformation is widely acknowledged, and CMA (CGH and SNP array) has been recommended for the first-tier analysis for these indications since 2010 [1,2]. CMA can sometime identify deletions which affect one single gene, helping to figure out whether haploinsufficiency truly drives pathological phenotypes [3,4].

Townes–Brocks syndrome (TBS, OMIM #104780) is an autosomal dominant condition, classically characterized by the triad of anorectal, ear, and thumb malformations described in 84%, 87% and 89% of affected patients, respectively [5]. Patients usually present with an imperforate anus, anal stenosis, or abnormal placement of the anus. Among ear anomalies, the most typical are overfolded superior helices, preauricular tags and microtia, frequently associated with hearing impairment (sensorineural and/or conductive). Limb malformations are represented by preaxial polydactyly, triphalangeal thumbs, hypoplastic thumbs, clubfoot, syndactyly of toes, or missing toes. Additional features are renal and urological anomalies (renal hypoplasia/dysplasia, multicystic kidneys, and vesicoureteral reflux), congenital heart defects (tetralogy of Fallot and ventricular septal defects), and genitourinary malformations. Developmental delay/intellectual disability (DD/ID) are also described in a subset of individuals. Overall, the clinical presentation of TBS is highly variable [6,7].

It is caused by mutations in *SALL1* [8], which encodes a zinc finger transcription factor exclusively found in the nucleus and linked to chromatin-mediated repression [9]. Recently, it was shown that beyond its transcriptional role, truncated SALL1 might impede the function of primary cilia [10].

The most common pathogenic variant c.826C>T (p.Arg276Ter) is detected in approximately half of simplex cases with TBS. Most (94%) individuals with this variant showed the characteristic triad of anal, thumb, and ear malformations, suggesting that it is associated with a more severe phenotype; furthermore, carriers of the p.Arg276Ter mutation seemed to have an increased risk of congenital heart defects [11]. Neither heterozygous nor homozygous Sall1 knockout mice show the TBS phenotype, but homozygotes die from kidney malformations such as those commonly seen in TBS [12]. Mouse mutants suggest that TBS is not caused by haploinsufficiency but could be due to expression of a dominant truncated *SALL1* protein that disrupts organ development [13]. With respect to the most common mutation reported in humans, Kiefer et al. (2008) proved that a truncated SALL1 protein is produced in B-cells from a TBS patient with p.Arg276Ter, showing that this variant escapes nonsense-mediated decay (NMD) and supporting the hypothesis that expression of a truncated protein is critical in the pathogenesis of TBS [14]. Nevertheless, a handful of patients with TBS harbor non-recurrent CNVs involving *SALL1*, showing that haploinsufficiency causes disease but may be associated with relatively mild phenotypes [15,16,17].

Here, we describe a family with a deletion which results in the loss of *SALL1* without the involvement of other genes; we review the phenotypes associated with all reported “*SALL1* only” deletions and with every patient known to carry the p.Arg276Ter mutation. We show a slight increase in congenital anomalies in patients with p.Arg276Ter, which is significant only for ear anomalies and congenital heart defects. We show the utility of CMA in the definition of atypical cases.

## 2. Materials and Methods

Informed consent forms for genetic analysis and publication of the results and clinical reports were signed by the patients or their parents in compliance with the national ethics regulation. DNAs were isolated from peripheral blood samples according to standard procedures. Array-CGH analyses were performed using CytoSure Oligo array ISCA v2 8 60 K OGT (Oxford Gene Technology, Oxford, UK), resolution ~150–210 kb. Protocols provided by the suppliers were followed without modification. Nucleotide designations were assigned according to the GRCh37/hg19 assembly of the human genome.

## 3. Results

### 3.1. Clinical Description

A 7-year-old girl was referred for clinical genetic evaluation for familiar hearing impairment. She was the only child of non-consanguineous parents, born at term with a weight of 3020 g, a length of 48 cm, an OFC of 32.3 cm, and an Apgar score of 6 and 10 at 1 and 5 min, respectively.

She presented with a ventrally positioned anus, pantonal sensorineural hearing loss (prevalent in the left ear) of mild–medium severity, and speech delay. At our evaluation, she was of slender build, had brachycephaly, protruding and asymmetric ears with low-set, and a posteriorly rotated and cupped right ear. She had cutaneous syndactyly between the 2nd and 3rd digit on both hands and feet. At the ophthalmologic examination, she had hypermetropia and tortuosity of the retinal vessels. Brain MRI did not show any morphological abnormality and the abdominal ultrasound was normal.

Her mother, maternal uncle, and maternal grandmother also had hearing impairments (Figure 1). The mother (II-2) had bilateral sensorineural hearing loss and asymmetrically positioned ears. She had no hand or feet abnormalities and no further clinical concerns, but she had three first-trimester miscarriages. The maternal uncle (II-3) had high frequency sensorineural hearing loss and an overfolded superior helix in the left ear and both hands showed brachydactyly with hypoplastic thumbs. At birth, he underwent surgery for anal stenosis. The abdominal and cardiac ultrasound were normal. The maternal grandmother had an hearing impairment but no known congenital anomalies; she died at 55.

### 3.2. Genetic Analysis

The proband underwent genetic tests for a hearing impairment. She was screened for GJB2- and GJB6-related hearing loss by sequencing of the entire GJB2 coding region and by searching for deletions D13S1830 and D13S1854 of GJB6; no mutations in the investigated genes were found. The study of mitochondrial DNA excluded the A1555G and A7445G mutations. Afterwards, next-generation sequencing for a panel of 62 genes related to hereditary hearing loss was conducted but no pathogenic mutations or variants of uncertain significance were identified.

Chromosomal analysis revealed a normal karyotype 46,XX. Finally, the girl was tested with array CGH, which identified the presence of a deletion on the long arm of chromosome 16 (del 16q12.1) of maternal origin. The deletion size was estimated to be 356 to 405 kb (chr16:51,179,338_51,535,461 NCBI Build 37), spanning exon 1 and the upstream region of SALL1 (Figure 2). The deleted region did not include other genes. The same deletion was present in the affected maternal uncle.

### 3.3. Review of Deletions in the Literature and in Databases

Distinct deletions encompassing SALL1 have been reported in the medical literature in six patients from four unrelated families, ranging in size from 3 to 600 kb approximately [15,16,17]. One further individual with a 10 kb SALL1 deletion and sufficient clinical details is present in the DECIPHER database. The clinical features are summarized in Table 1 and the deletions are shown in Figure 2.

Briefly, Borozdin et al. described three families with SALL1 deletion identified by quantitative PCR, in a cohort of patients where SALL1 sequencing had not identified point mutations [15]. We do not report family 2 here because the deletion, of approximately 2.5 Mb, affects not only SALL1 but other genes as well. In family 1, two siblings were reported: a female with bilateral sensorineural hearing loss, a dysplastic left ear, a ventrally positioned anus, bilateral triphalangeal thumbs, and metatarsus adductus, and her brother with a slightly dysplastic right ear, membranous anal atresia at birth, and broad thumbs. Their father could not be examined but was reported to have bilateral hearing loss. Both siblings had a heterozygous 75 kb deletion on chromosome 16q12.1, spanning exons 2 and 3 and a large part of intron 1 of SALL1. In family 3, a female patient with a negative family history had bilateral hearing loss, protruding small ears, a preauricular pit, an ectopic and narrow anus, hammer toes and hallux valgus on the right side, relatively broad thumbs, and vesicoureteral reflux that caused reflux nephropathy and ultimately renal failure. She was found to have a 3384 bp deletion on chromosome 16q12.1, spanning part of exon 2 and 3 and intron 2 of SALL1.

Miller et al. described a family harbouring a 149 kb deletion encompassing SALL1 but no other genes, identified by CMA [16]. The proband had a simple, cupped right ear, an imperforate anus, brachydactyly with normal thumbs, small feet with thick toes, short halluces bilaterally, an incurved left great toe, hypotonia, ocular coloboma, and developmental delay. He inherited the deletion from his father, who was born with an imperforate anus, had learning difficulties in school, and presented with preaxial polydactyly of the right hand.

Stevens and May reported on a patient with a 605 kb deletion upstream of SALL1, detected by CMA, resulting in the possible loss of a critical cis-regulatory region [17]. The patient’s phenotype was characterized by an asymmetric ear size, an imperforate anus with rectovaginal fistula, mildly broad thumbs and fingertips, behavioral and cognitive difficulties, renal dysfunction, vesicoureteral reflux, lipoma at the filum terminale, joint laxity, hypotonia, and poor balance. Parental samples were not available. The author proposed that the phenotype could be due to the loss of SALL1 cis-regulatory elements.

In the DECIPHER database [18], a 10.71 kb SALL1 deletion was reported in a patient with bilateral sensorineural hearing impairment, anal stenosis, broad fingertips, mild global developmental delay, and penile hypospadias.

Overall, among the ten patients with SALL1 deletions, seven had an imperforate anus or anal stenosis and two had a ventrally positioned anus. Dysplastic ears were reported in six patients, and one had an asymmetric ear size. Regarding hand malformations, one had triphalangeal thumbs, one had preaxial polydactyly, one had hypoplastic thumbs, four had broad thumbs and/or fingertips, and one had syndactyly. Three patients also had feet malformations. Hearing impairment was present in eight patients and vesicoureteral reflux was present in two. None had congenital heart disease. Four patients showed developmental delay or intellectual disability, generally reported to be mild. Additional features were penile hypospadia (1 patient), hypotonia (2), and coloboma (1).

### 3.4. Review of Clinical Features in Carriers of the p.Arg276Ter Mutation and Comparison with Deletion Carriers

Afterwards, we reviewed the clinical features of patients carrying the recurrent SALL1 p.Arg276Ter mutation, which was shown to escape NMD and is expected to cause disease with a dominant-negative mechanism. To our knowledge, 20 patients have been reported since 1999 and their clinical features are summarized in Table 2 [11,19,20,21,22,23,24,25].

We compared the prevalence of each clinical feature in two groups of patients, the first one being represented by patients with SALL1 deletions (“haploinsufficiency” group) and the second one by patients with the p.Arg276Ter mutation (dominant-negative). The percentage of patients with a particular phenotypic characteristic is represented graphically in Figure 3, with the first column showing the results as reported on GeneReviews, which is expected to represent the global burden of mutations, whereas the following columns show the features in the haploinsufficiency and dominant-negative groups.

A Fisher’s exact text was used to compare each clinical feature’s prevalence between the groups (Table 3). Anal malformations are equally represented (70% in group 1 and 68% in group 2), whereas the other anomalies appear to be more frequent in individuals with dominant-negative mutations, although only “dysplastic ears” and “congenital heart defects” reach a statistically significant difference (*p*-value = 0.0230 and 0.0095, respectively).

Ear abnormalities are described in 6 out of 9 patients with SALL1 deletion, while they are reported in all 20 patients with the p.Arg276Ter mutation.

Congenital heart defects are not reported in any patients with SALL1 deletion, while they are present in 55% of patients with the p.Arg276Ter mutation (six patients had ventricular septum defects, two had tetralogy of Fallot, one had an atrial and ventricular septum defect, and one had patent foramen ovale and bidirectional patent ductus arteriosus) [5,11].

## 4. Discussion

TBS has specific clinical diagnostic criteria, but the widespread availability of sequencing and CMA has allowed for the identification of atypical cases. In evaluating genotype–phenotype correlations, large deletions and point mutations introducing premature stop-codons are usually associated with haploinsufficiency. TBS is an exception to this rule, as the most common *SALL1* mutation (p.Arg276Ter) was found to escape NMD [13] and it acts through a dominant-negative mechanism. Deletions that encompass *SALL1* and do not include other genes were initially associated with milder phenotypes of TBS when compared with dominant-negative mutations [15], but genotype–phenotype correlations were later debated [16].

An investigation of mutational mechanisms was performed for a few more *SALL1* mutations; a patient who presented with isolated right hand preaxial polydactyly and no family history of congenital malformation had a *SALL1* frameshift mutation 3414_3415delAT [26]. The phenotype was mild, and the authors showed that the mutation was indeed subject to NMD. Interestingly, the same 3414_3415delAT mutation in *SALL1* had also been described in a 17-year-old patient with bilateral renal hypodysplasia without any extra-renal manifestations [27]. Furniss et al. also studied a distinct *SALL1* point mutation, 995delC, identified in a patient with typical TBS (preaxial polydactyly, an imperforate anus, rectal atresia, hypospadias, and overfolded helices) and showed that it was not subject to NMD [26]. These results add up to the notion that mutations leading to haploinsufficiency (large deletions and point mutations subject to NMD) globally cause milder phenotypes, with respect to dominant-negative *SALL1* mutations.

The scenario becomes more complex when we consider that NMD was investigated in a unique “recessive” *SALL1* mutation, c.3160C>T (p.R1054*), identified in a homozygous state in two siblings affected by severe ID and multiple congenital anomalies (affecting the brain, heart, ears, limbs, and kidneys) [28]. In this family, 11 mutation carriers did not show overt clinical signs which could suggest TBS. A mutant transcript was found to partially undergo NMD, indicating that a decreased amount of mutant protein is produced. The truncated protein resulting from p.R1054* must thus preserve some degree of SALL1 function, apparently sufficient to prevent heterozygotes from showing TBS features.

Considering that the wild-type SALL1 polypeptide consists of 1324 amino acids, truncating mutations which lie in the distal 3′ end of SALL1 such as p.R1054* may cause milder effects or no effects (at the heterozygous state) even if they do not undergo NMD, given the retention of functional protein domains.

Although more mutations should be functionally characterized in order to confirm these findings, a growing degree of severity can be hypothesized such as: (1) the complete absence of *SALL1* is likely to be embryonic lethal, (2) a small amount of SALL1 function (with the homozygous p.R1054* mutation being the only example to date) causes severe developmental disorder with onset in foetal life, (3) haploinsufficiency (large deletions and mutations which do not escape NMD, such as 3414_3415delAT) causes mild TBS, and 4) dominant-negative mutations (such as the recurrent c.826C>T (p.Arg276Ter) and 995delC) causes typical TBS. The genetic background will further add to the complexity of this picture and would partly explain intrafamilial variability.

The major limit of this study is that large deletions involving *SALL1* are still very limited in number, and only a few mutations were tested for NMD. Furthermore, no Western blot experiments were performed to formally test protein reduction levels. Nevertheless, congenital anomalies (except for dysplastic ears) seem to be less prevalent and less severe in patients with haploinsufficiency. At least three deletions, including those in the present report, were identified in families which were not tested for TBS as a first hypothesis, confirming that the full spectrum of this condition was not present, although significant anomalies could be evaluated retrospectively.

The phenotypic spectrum of TBS represents a continuum and the distinction between mild and severe cases is not straightforward. According to the clinical criteria, TBS diagnosis is established in a proband with three major features (an imperforate anus or anal stenosis, dysplastic ears, and typical hand malformations). If only two major features are present, the presence of minor features, such as a hearing impairment, renal anomalies, foot malformations, and congenital heart defects, supports the diagnosis [5]. The complete triad of TBS’s major clinical signs was present in only 3 out of 10 patients with *SALL1* deletions (group 1: 30%) and in 12 out of 20 patients with the p.Arg276Ter mutation (group 2: 60%). Although the sample is small, it appears that the patients in group 1 show a less characteristic TBS phenotype, which makes clinical diagnosis more challenging.

Renal abnormalities are also less frequent than expected in patients with deletion-associated TBS, although the difference does not reach statistical significance (*p*-value = 0.2087).

Congenital heart defects are reported in 50% of TBS patients with the common p.Arg276Ter mutation and 12–25% of individuals with other *SALL1* pathogenic variants [5]. Reviewing the literature on deletions involving *SALL1*, not a single subject was reported to have a congenital heart defect. Despite the small sample size, this feature is clearly uncommon among patients with *SALL1* deletions.

Altogether, these findings suggest that patients with *SALL1* deletions show a milder phenotype, since they are less likely to fulfil the three major diagnostic criteria and have a lower frequency of renal and cardiac disease.

Larger datasets will also inform on the true prevalence of DD/ID. A slight increase in cognitive dysfunction in patients with large deletions including *SALL1* may represent a bias, since CMA is usually recommended in patients with developmental disorders; on the other hand, a long-range effect on the expression of neighboring genes may be possible. CMA has already been shown to be useful in patients with hearing impairments [29] and should be strongly indicated if hearing loss is associated with congenital anomalies.

Considering SALL1’s role in chromatin-mediated repression [9], future studies may address DNA methylation profiles in order to assess if a specific episignature is present for this disorder [30] and if it is different among patients with haploinsufficiency and those with dominant-negative mutations.

## 5. Conclusions

We provide further evidence that an abnormal truncated *SALL1* protein, acting in a dominant-negative fashion, is responsible for the full spectrum of developmental defects seen in TBS, whereas *SALL1* haploinsufficiency causes a milder TBS-like phenotype.

## Figures and Tables

**Figure 1 genes-14-00258-f001:**
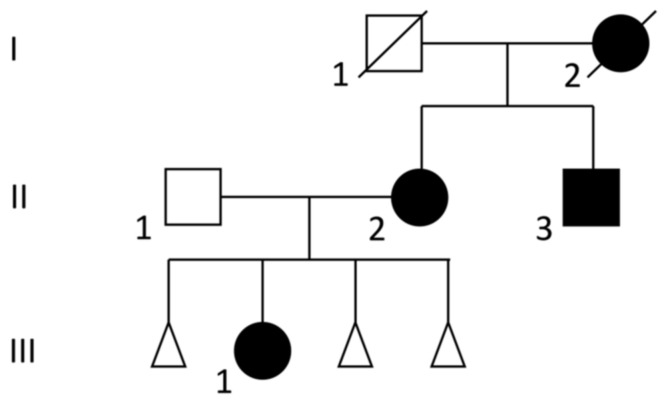
Pedigree of the family. Full symbols indicate the affected family members. Circles represent female family members, squares male family members, triangles miscarriages.

**Figure 2 genes-14-00258-f002:**
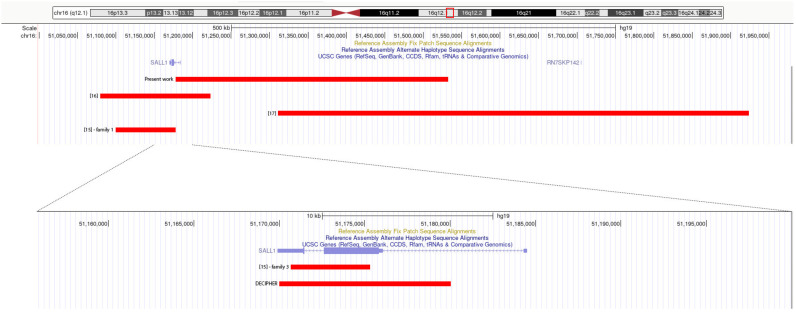
Physical map of the 16q12.1 region (51,000,000 to 52,000,000, GRCh37/hg19) adapted from the UCSC genome browser. Red box indicates the region of interest on chromosome 16. Red bars indicate the genomic regions involved in the family reported here and in those from the literature [15,16,17] and the DECIPHER database.

**Figure 3 genes-14-00258-f003:**
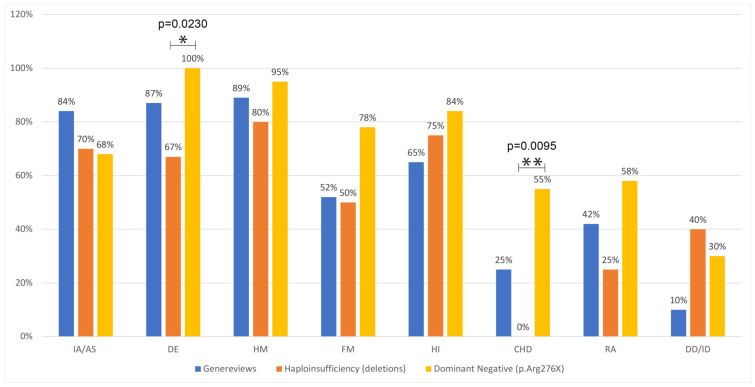
Comparison among the prevalence of diverse clinical features as reported in GeneReviews, in patients with *SALL1* deletions and with the p.Arg76Ter mutation. AT: anal tag; CHD: congenital heart defects; DD/ID: developmental delay/intellectual disability; DE: dysplastic ears; FM: feet malformations; HI: hearing impairment; HM: hand malformations; IA/AS: imperforate anus/anal atresia; RA: renal anomalies; VPA: ventrally positioned anus.

**Table 1 genes-14-00258-t001:** Clinical features in patients with deletions encompassing SALL1. AES: asymmetric ear size; CHD: congenital heart defects; DD/ID: developmental delay/intellectual disability; DE: dysplastic ears; FM: feet malformations; HI: hearing impairment; HM: hand malformations; IA/AS: imperforate anus/anal atresia; RA: renal anomalies; VPA: ventrally positioned anus. Highlighted in light green are patients showing the complete classical triad of TBS-related malformations (anorectal, ear, and hand malformations).

Deletions	Family	Patient	IA/AS	DE	HM	FM	HI	CHD	RA	DD/ID
This report	1	1	VPA	+	+	+	+	−	−	+
		2	−	−	−	−	+	−	−	−
		3	+	+	+	−	+	−	−	−
[15]	1	1	VPA	+	+	+	+	NR	NR	−
		2	+	+	+	−	NR	−	−	−
	3	1	+	+	+	+	+	−	+	−
[16]	1	1	+	+	−	+	−	−	−	+
		2	+	−	+	−	−	−	−	−
[17]	1	1	+	AES	+	NR	NR	−	+	+
DECIPHER	1	1	+	NR	+	NR	+	NR	NR	+
TOT	6	10	7/10	6/9	8/10	4/8	6/8	0/8	2/8	4/10

**Table 2 genes-14-00258-t002:** Clinical features in patients with the SALL1 p.Arg276Ter mutation. AT: anal tag. Highlighted in light green are patients showing the complete classical triad of TBS-related malformations (anorectal, ear, and hand malformations).

p.Arg276Ter	Patient	IA/AS	DE	HM	FM	HI	CHD	RA	DD/ID
[19]	3-1	+	+	+	+	+	−	−	+
	4-1	+	+	+	+	+	−	+	−
	5-1	−	+	+	+	+	+	+	−
[20]	TB6	+	+	+	−	+	+	−	NR
	TB7	+	+	+	−	+	−	+	NR
	TB8	+	+	+	+	+	−	+	NR
[21]	2	AT	+	+	+	NR	+	+	NR
[11]	103/1	AT	+	+	+	+	+	+	−
	119/1	+	+	+	+	+	−	−	NR
	149/1	+	+	+	−	+	+	+	NR
	182/1	+	+	+	+	−	−	−	+
	186/1	+	+	+	+	+	+	−	−
	188/1	+	+	−	−	−	−	+	−
	205/1	AT	+	+	+	+	NR	+	−
	224/1	VPA	+	+	+	+	+	−	−
	224/2	+	+	+	+	−	+	−	NR
[22]		AT	+	+	+	+	NR	−	NR
[23]		+	+	+	+	+	−	+	+
[24]	B934	NR	+	+	NR	+	+	NR	NR
[25]		+	+	+	NR	+	+	+	NR
TOT	20	13/19	20/20	19/20	14/18	16/19	10/18	11/19	3/10

**Table 3 genes-14-00258-t003:** Prevalence of diverse clinical features in patients with SALL1 deletions (group 1) and with the common p.Arg276Ter mutation (group 2). A two-tailed Fisher’s exact test was used to determine if there was a significant association between the type of SALL1 mutation (group 1 vs. group 2) and the presence/absence of each clinical feature.

	IA/AS		DE		HM		FM		HI		CHD		RA		DD/ID	
	Yes/Tot	%	Yes/Tot	%	Yes/Tot	%	Yes/Tot	%	Yes/Tot	%	Yes/Tot	%	Yes/Tot	%	Yes/Tot	%
Group 1: haploinsufficiency (deletions)	7/10	70%	6/9	67%	8/10	80%	4/8	50%	6/8	75%	0/8	0%	2/8	25%	4/10	40%
Group 2: dominant negative (p.Arg276Ter)	13/19	68%	20/20	100%	19/20	95%	14/18	78%	16/19	84%	10/18	55%	11/19	58%	3/10	30%
*p*-value	1.0000		0.0230		0.2512		0.1972		0.6159		0.0095		0.2087		1.0000	

## Data Availability

The data that support the findings of this study are available from the corresponding author upon reasonable request.

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
