# Peer review of "Chromosomal Microarray Analysis Identifies a Novel SALL1 Deletion, Supporting the Association of Haploinsufficiency with a Mild Phenotype of Townes–Brocks Syndrome"

_genes, 2023, doi:10.3390/genes14020258_

Round 1

Reviewer 1 Report

Comments to Authors:

The manuscript ‘Chromosomal microarray analysis identifies a novel SALL1 deletion, supporting the association of haploinsufficiency with a mild phenotype of Townes-Brocks syndrome’ by AM Innoceta and colleagues describes the identification of a new deletion in the SALL1 gene.   

1.       It would be helpful to further clarify ‘mild phenotype’ as repeatedly indicated in the title and throughout the text with specification by organ phenotype (or, for at least some of the affected organs).

2.   Could additional description of the congenital heart disease in patients with Townes-Brocks syndrome with the p.Arg276Ter variant be provided.  Do patients with other variants have congenital heart disease; how is this a reflection of a ‘more severe’ phenotype, or could this feature just be different/specific for the p.Arg276Ter variant?

3.   The wild type SALL1 polypeptide consists of 1324 amino acids.  In the text discussion pertaining to effects of gain of function versus haploinsufficiency of the SALL1 gene, it would be helpful to clarify that there may be gain of function differences with short remaining polypeptides (such as only 275 amino acids with p.Arg276Ter, or of 322, 371, 418 amino acids as listed at OMIM) versus others such as polypeptides of 1053 amino acids (p.Arg1054Ter) given the marked differences in residual protein domains.   

4.   Referring to the data presented in Figure 3, what was the p-value for the comparison of renal anomalies between cases with haploinsufficiency cases versus those with p.Arg276Ter?

5.   Note that the legends of Tables 1 and 3, and Figure 3 appear truncated with the boxed presentation, they should be corrected.  The abbreviations if used consistently, would only have to be listed once. 

Minor: Check spelling and grammar on lines 232, 249, 250 of page 8/10. 

Author Response

I thank the reviewer for comments and suggestions.

  1. It would be helpful to further clarify ‘mild phenotype’ as repeatedly indicated in the title and throughout the text with specification by organ phenotype (or, for at least some of the affected organs). This is a crucial issue and we tried to make it clearer in the discussion, where we provide further description of the phenotype: we underlie that the phenotypic spectrum of TBS represents a continuum, but the presence of the three major diagnostic criteria is more frequent in p.Arg276Ter carriers, who also show a higher incidence of heart defect and renal disease.
  2.  Could additional description of the congenital heart disease in patients with Townes-Brocks syndrome with the p.Arg276Ter variant be provided.  Do patients with other variants have congenital heart disease; how is this a reflection of a ‘more severe’ phenotype, or could this feature just be different/specific for the p.Arg276Ter variant? We provided a description of the congenital heart defects presented by patients with Townes-Brocks syndrome with the p.Arg276Ter variant. We report that 12%-25% of individuals with TBS and a SALL1 pathogenic variant other than p.Arg276Ter have congenital heart defects too, whereas no large deletions carriers show cardiac abnormalities to our knowledge.
  3.  The wild type SALL1 polypeptide consists of 1324 amino acids.  In the text discussion pertaining to effects of gain of function versus haploinsufficiency of the SALL1 gene, it would be helpful to clarify that there may be gain of function differences with short remaining polypeptides (such as only 275 amino acids with p.Arg276Ter, or of 322, 371, 418 amino acids as listed at OMIM) versus others such as polypeptides of 1053 amino acids (p.Arg1054Ter) given the marked differences in residual protein domains. We added this further issue in the discussion regarding the interpretation of different variants. 
  4.  Referring to the data presented in Figure 3, what was the p-value for the comparison of renal anomalies between cases with haploinsufficiency cases versus those with p.Arg276Ter? Figure 3 was split in Figure 3 and Table 3 in order to include the p-values in the table. In Figure 3 we added the actual p values above the *. We added p values for the non-significant groups in Table 3.
  5.  Note that the legends of Tables 1 and 3, and Figure 3 appear truncated with the boxed presentation, they should be corrected.  The abbreviations if used consistently, would only have to be listed once.The format of legends was modified in all figures and tables and abbreviations were deleted where already listed.

Minor: Check spelling and grammar on lines 232, 249, 250 of page 8/10.  We modified this. Thank you!

Reviewer 2 Report

This manuscript describes a novel deletion in SALL1 and compares the phenotypes between haploinsufficient and dominant-negative TBS patients. There are some points that the authors should address:

1. For Figure3, please show actual p values instead of *. Also the authors should show p values for the non-significant groups as well.

2. The authors claim that the deletion patients are haploinsufficiency, which could be true, but this is weakly supported by real data. Did the authors actually examine the SALL1 protein levels e.g. using western blot to see if there is a 50% reduction in protein levels? Similarly, did the authors sequence the WT copy of SALL1 to make sure there is no mutations, althought this would be a very rare case. This would be key to make the conclusion that the deletion patients, at least the novel patient in this study, is actually haploinsufficiency.

3. Table legends are truncated in the current manuscript. 

Author Response

We thank the reviewer for comments and suggestions.

  1. For Figure3, please show actual p values instead of *. Also the authors should show p values for the non-significant groups as well. Figure 3 was split in Figure 3 and Table 3 in order to include in the table the p-values. In Figure 3 we added the actual p values above the *. We added p values for the non-significant groups in Table 3
  2. The authors claim that the deletion patients are haploinsufficiency, which could be true, but this is weakly supported by real data. Did the authors actually examine the SALL1 protein levels e.g. using western blot to see if there is a 50% reduction in protein levels? Similarly, did the authors sequence the WT copy of SALL1 to make sure there is no mutations, althought this would be a very rare case. This would be key to make the conclusion that the deletion patients, at least the novel patient in this study, is actually haploinsufficiency. We did not sequence SALL1, because the rare patients who carry mutations on both alleles of SALL1 show a much more severe phenotype. We could not perform western blot nor rt-PCR on SALL1 mRNA to formally prove a 50% reduction in protein or mRNA levels. We think this experiment would be especially interesting for the deletion reported by Stevens and May, which does not affect SALL1 coding region directly. We report a deletion that involves the whole gene promoter region and exon 1 and it is hard to believe that a protein will be translated. Still, although the functional effect of deletions is usually straightforward, it is true that this was not formally tested, and the level of translation of a single copy of the gene could be modulated by several factors and differ among individuals. We added among the limits of this study that “Furthermore, no western blot experiments were performed to formally test protein reduction levels”.

3. Table legends are truncated in the current manuscript. We apologize and hope that table legends are now in a correct position.